physical chemistry

layered oxyselenides, solid-state reaction, band structure

**Authors for correspondence:**
Shugang Tan
e-mail: tanshugang@sdut.edu.cn
Yuping Sun
e-mail: sunyuping@sdut.edu.cn

This article has been edited by the Royal Society of Chemistry, including the commissioning, peer review process and editorial aspects up to the point of acceptance.

# Synthesis, structure and physical properties of the new layered oxyselenides Bi$_2$LnO$_4$Cu$_2$Se$_2$ (Ln = rare earth)

Shugang Tan[1], Chenhao Gao[1], Cao Wang[1], Qiang Jing[1], Tong Zhou[1], Guangchao Yin[1], Meiling Sun[1], Fei Xing[1], Rui Cao[2] and Yuping Sun[1]

[1]School of Physics and Optoelectronic Engineering, and [2]Office of International Cooperation and Exchange, Shandong University of Technology, Zibo 255000, People's Republic of China

We have synthesized a new series of layered oxyselenides Bi$_2$LnO$_4$Cu$_2$Se$_2$ (Ln=Nd, Sm, Eu, Dy, Er, Yb). Their crystal structures and physical properties were studied through X-ray diffraction, electric transport measurements, bulk magnetization and first-principle calculation. All these compounds have a tetragonal structure with space group I4/mmm. They exhibit hole-type metallic behaviours which is also verified by the DFT calculation. The new Bi$_2$LnO$_4$-type block in these compounds may give people some enlightenment in synthesizing new iron-based superconductors or other layered compounds.

## 1. Introduction

Since the discovery of superconductivity, people have been trying to find new superconductors with application values. Compounds with layered structure have always attracted people's attention because of their rich properties. Especially in recent years, many layered structural compounds have been found to exhibit superconductivity, such as cuprates superconductors [1], iron-based superconductors [2] and BiS$_2$-based superconductors [3]. It has become one of the most important ways to search for new superconductors by exploring novel layered structural compounds.

Oxychalcogenides tend to adopt layered structure, due to the different sizes and coordination requirements of the oxide and the heavier chalcogenide anions [4]. Oxychalcogenides have been extensively studied because of their novel electronic or magnetic properties and intriguing structure features. Generally,

TMCh-based (TM = Cu, Ag; Ch = S, Se, Te) oxychalcogenides dominated the known chemistry of oxychalcogenides, which possess very diverse and interesting structures and exhibit useful physical and chemical properties [4–9]. LnOTMCh (Ln = lanthanide) is a typical class of layered oxychalcogenides. The formulation as LaOCuS emphasizes the occurrence of two fairly distinct layer types, formally $[LaO]^+$ and $[CuS]^+$. The analogue BiCuSeO has attracted much attention in the thermoelectric field [10–13]. Another important type is represented by the $Sr_2MO_2Cu_2Ch_2$, the structure of which was first described for the oxide antimonide $Sr_2Mn_3Sb_2O_2$ [14–20]. These layered types have great flexibility and are each merely the most common members of the structural homologous series. In 2008, the discovery of Fe-based superconductor opens a new chapter in the research of high-temperature superconductivity. The common feature of iron-based superconductors is that they have $Fe_2Pn_2$ antifluorite layer. Many of the compounds with CuSe layers have analogues with FeAs layers, and there could be iron arsenides with this structure. In 2002, Evans *et al.* reported a new family of layered oxyselenide $Bi_2LnO_4Cu_2Se_2$, and five compounds were synthesized (Ln = Y, Gd, Sm, Nd, La) [21]. However, the physical properties of this series of oxyselenide are still unstudied. Recently, we studied the physical and electrical properties of $Bi_2YO_4Cu_2Se_2$, which exhibits quasi-two-dimensional metallic behaviour [22]. More compounds are needed to enrich this class of materials.

In this work, we report the synthesis, structure and physical properties several new compounds of the series $Bi_2LnO_4Cu_2Se_2$ (Ln = Sm, Nd, Eu, Dy, Er, Yb), which could give people some enlightenment in synthesizing new iron-based superconductors or other layered compounds.

## 2. Experimental details

$Bi_2LnO_4Cu_2Se_2$ sample was prepared by reacting a stoichiometric mixture of $Bi_2O_3$, $Ln_2O_3$, Bi, Cu and Se. The chemical equation can be written as

$$5Bi_2O_3 + 3Ln_2O_3 + 2Bi + 12Cu + 12Se \rightarrow 6Bi_2LnO_4Cu_2Se_2.$$

The raw materials were mixed and ground thoroughly in an agate pestle and mortar, and then the mixture was pressed into pellets under 12 MPa. The pellets were placed into dried alumina crucibles and sealed under vacuum (less than $10^{-4}$ Pa) in the silica tubes which had been baked in a dry box for 1–2 h at 150°C. The ampoules were heated to 830°C with 1°C min$^{-1}$ and maintained at this temperature for 24 h. Finally, the furnace was shut down and cooled to room temperature naturally. The obtained samples were reground, pelletized and heated for another 24 h at 830°C followed by furnace cooling. The X-ray powder diffraction patterns were recorded at room temperature on a Panalytical diffractometer (X'Pert PRO MRD) with Cu K$\alpha$ radiation (40 kV, 40 mA) and a graphite monochromator in a reflection mode ($2\theta = 10$–$90°$, step = $0.016°$, scan speed = 5 s per step). Structural refinement of the samples was carried out by using Rietica software. Magnetic susceptibility measurements were carried out using a Quantum Design MPMS5 magnetometer in the temperature range 5–300 K. Approximately 20–40 mg of material was weighed accurately into a gelatin. Measurements were made on warming in a field of 1 kOe, first after cooling in zero field (ZFC) and then again after cooling in the measuring field (FC). The electrical resistivity and thermoelectric property were measured using a Quantum Design physical properties measurement system (PPMS). The electric structure was obtained from first-principles density functional theory (DFT) in the generalized gradient approximation (GGA) according to Perdew *et al.* [23], which were calculated using the plane-wave projector augmented method as implement in the Vienna *ab initio* simulation package (VASP). An energy cut-off of 520 eV was used. The convergence criterion energy was set to be $10^{-6}$ eV per unit cell and the forces on all relaxed atoms were less than 0.01 eV Å$^{-1}$.

## 3. Results and discussion

The structure of $Bi_2LnO_4Cu_2Se_2$ is shown in figure 1, which can be described as stacking of edge-shared $Cu_2Se_2$ tetrahedron layers with $Bi_2LnO_4$ layers alternatively along the c-axis. The $Bi_2LnO_4$ layer can be described as '$[M_3O_4]^{+}$' layer, which is a double fluorite-type slab. The $[M_3O_4]$ units have been observed in several series of copper oxide superconductors, for example $(Ce,Ln)_3Sr_2Cu_3O_{11}$ and $(Y,Ce)_3SrCuFeO_9$ [24–26]. On the other hand, this type of oxyselenides has a similar structure to the oxyhalides. The structure of BiOCuSe is derived from that of PbFCl or BiOCl through the replacement of chloride by selenide and the incorporation of the $Cu^+$ ions into tetrahedral sites coordinated by selenide ions. Thus, the ZrSiCuAs structure of La(Bi)OCuS(Se) is also often described as the stuffed or

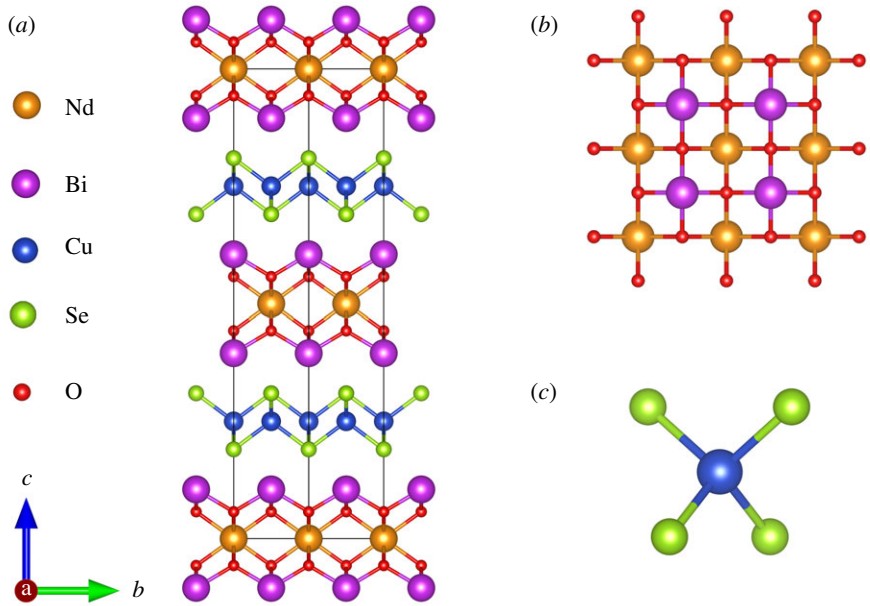

**Figure 1.** (*a*) The crystal structure of $Bi_2NdO_4Cu_2Se_2$; (*b*) the $Bi_2LnO_4$ layers from a vertical view; (*c*) the $Cu_2Se_2$ tetrahedral structure.

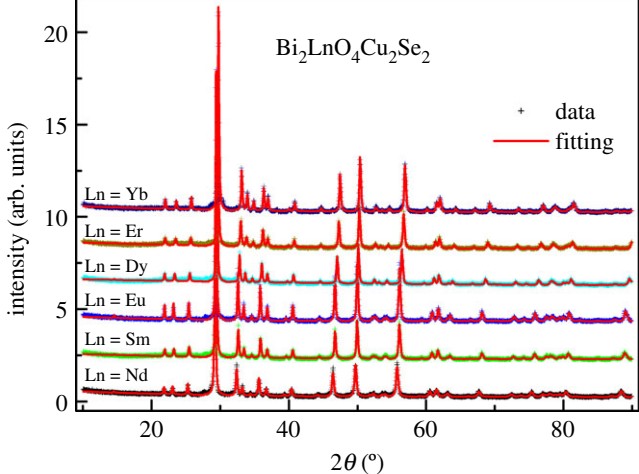

**Figure 2.** The XRD patterns of $Bi_2LnO_4Cu_2Se_2$.

filled PbFCl structure. The relation between $Bi_2NdO_4Cl$ and $Bi_2NdO_4Cu_2Se_2$ is different from that of BiOCl and BiOCuSe (electronic supplementary material, figure S1). In fact, the unit cell of $Bi_2NdO_4Cu_2Se_2$ can be regarded as double units of BiOCuSe, which are connected by the vertical Bi-site atoms but placed in opposite directions along c-axis. Figure 1*b* shows the $Bi_2LnO_4$ layers from a vertical view. The Ln ions located in square-planar environments in this compound, which provide a model of two-dimensional physical properties of the rear earth element. Figure 1*c* shows the $Cu_2Se_2$ tetrahedral structure.

We successfully synthesized a series of $Bi_2LnO_4Cu_2Se_2$ (Ln = Nd, Sm, Eu, Dy, Er, Yb), in which the Nd, Sm analogues were first synthesized by Evans *et al.* [21] and the Eu, Dy, Er, Yb analogues are reported for the first time. All the samples are black, which is consistent with their electric transport properties in the following. Figure 2 shows the powder X-ray diffraction (XRD) patterns of $Bi_2LnO_4Cu_2Se_2$, scanning over a $2\theta$ range of 10–90° at room temperature. Using the tetragonal structure with the space group of I4/mmm, the XRD patterns for the whole compounds can be fitted very well, which are shown with red lines in figure 2. (The details are shown in the electronic supplementary material.) With the Ln ion changing from Nd to Yb, the diffraction peaks gradually shift to higher angle degree, suggesting a contraction of the lattice. The results obtained from the refinement of the X-ray diffraction data are shown

**Table 1.** Fractional atomic coordinates of $Bi_2LnO_4Cu_2Se_2$.

| atom | x | y | z |
| --- | --- | --- | --- |
| Bi | 0.5 | 0.5 | z(Bi) |
| Ln | 0.5 | 0.5 | 0.5 |
| O | 0 | 0.5 | z(O) |
| Cu | 0 | 0.5 | 0.25 |
| space group | I4/mmm | | |

in tables 1–3. It can be seen that when the Ln changes from Nd to Yb, the lattice parameters a and c decrease correspondingly, which can be explained by the decrease of the ionic sizes of Ln. Bond lengths provide information about the nature of the chemical bonds. The measured bond length (d) of an ideal ionic bond is close to the estimated bond length given by the sum of the ionic radii of a cation ($d_c$) and a neighbouring anion ($d_a$). The length of the Cu-Se bond was nearly equal in $Bi_2LnO_4Cu_2Se_2$ ranging from 2.42 to 2.45 Å. But the length of Cu-Se bond is shorter than that in BiOCuSe (2.51 Å) and LaOCuSe (2.52 Å) [27]. The Cu ion in the semiconducting BiOCuSe and LaOCuSe is monovalent. The Cu in $Bi_2LnO_4Cu_2Se_2$ should be in a higher valence state than in BiOCuSe, because the ionic radii of $Cu^{2+}$ is shorter than that of $Cu^+$. The bond length indicates that the Cu cation provides more electrons to the neighbouring Se anion. Our previous results have indicated the existence of a mixed-valence state of $Cu^{2+}/Cu^+$ in $Bi_2YO_4Cu_2Se_2$, which causes the metallic behaviour in this compound. The Bi-O bond lengths in $Bi_2LnO_4Cu_2Se_2$ are shorter than that in BiOCuSe (2.33 Å) [27], which suggests the Bi-O has greater covalent character. On the contrary, the Bi-Se bond in $Bi_2LnO_4Cu_2Se_2$ lengths are longer than that in BiOCuSe (3.23 Å) [27], indicating the Bi-Se bond are weaker. These observations indicate the two-dimensional nature of $Bi_2LnO_4Cu_2Se_2$ is stronger than that in BiOCuSe.

Recently, we studied the physical properties of the analogue $Bi_2YO_4Cu_2Se_2$. The theoretical calculation indicated that the ground state of this compound is the quasi-two-dimensional metal state. The states of the valence band maximum are mainly composed of antibonding Cu-3d/Se-4p states and the conduction band are mainly Bi-6p/O-2p states. When the transferred valence electrons from the blocking layer to $Cu_2Se_2$ layer are less than two per layer, the compound would exhibit metallic behaviours. The electric transport properties of $Bi_2LnO_4Cu_2Se_2$ are shown in figure 3. All these compounds show metallic behaviours in the measured temperature region, which agrees with the conclusion we obtained from theoretical calculations. For all the samples, the resistance drops linearly from 300 to 100 K. The influence of Ln atoms to the transport properties is mainly embodied in the influence of the structure of the Cu-Se layer. As the Ln atom changed from Nd to Yb, the size of which is decreasing, the angle of Se-Cu-Se and the distance of Cu-Cu tend to decrease. As a result, the electrons transferred easily in Cu-Se layers. This can be demonstrated by the decreasing resistivity and smaller temperature coefficient of resistivity as Ln changing from Nd to Yb. At low-temperature region, the resistance is satisfied with the Fermi-liquid behaviour in the ground state and we fitted the resistance curve using the equation $\rho = \rho_0 + AT^2$, where $\rho_0$ is the residual resistivity, and the coefficient A represents the inelastic scattering between electrons and is generally proportional to the square of the effective electron mass. (The values of the fitted $\rho_0$ and A are shown in electronic supplementary material, table S1.) We also measured the thermoelectric transport properties of all the samples. The Seebeck coefficients are positive in the whole measured temperature region, indicating the major carriers are holes for all the samples. (The measured Seebeck coefficient S, thermal conductivity $\kappa$, electrical conductivity $\sigma$ and figure of merit ZT (ZT = $S^2T\sigma/\kappa$) at room temperature are shown in electronic supplementary material, table S2.)

We measured the magnetic properties of the samples. Plots of the temperature dependence of the magnetic susceptibility for $Bi_2LnO_4Cu_2Se_2$ (Ln = Sm, Nd, Eu, Dy, Er, Yb) are shown in electronic supplementary material, figure S3. These samples have relatively high susceptibility compared with $Bi_2YO_4Cu_2Se_2$. The order of susceptibility from large to small is Dy, Er, Yb, Nd and Sm, which is related to the occupation of the 4f orbital in the Ln ions. All the measured samples exhibit paramagnetic behaviour in the measured temperature region from 5 to 300 K. For Dy and Eu analogues, the inverse susceptibility curves are relatively linear with temperatures nearly in the whole measured temperature region, which could be fitted by the normal Curie–Weiss law, $\chi = C/(T - \theta_p)$. For Nd and Yb analogues, their susceptibility curves obey the Curie–Weiss law only in the higher temperature regions, and the inverse susceptibility curves deviate from linearity at low temperatures.

**Table 2.** Results of powder X-ray diffraction and Rietveld refinements for $Bi_2LnO_4Cu_2Se_2$ at room temperature.

| formula | $Bi_2NdO_4Cu_2Se_2$ | $Bi_2SmO_4Cu_2Se_2$ | $Bi_2EuO_4Cu2Se_2$ | $Bi_2DyO_4Cu_2Se_2$ | $Bi_2ErO_4Cu_2Se_2$ | $Bi_2YbO_4Cu_2Se_2$ |
|---|---|---|---|---|---|---|
| molecular weight | 911.210 | 917.330 | 918.934 | 929.470 | 934.230 | 940.010 |
| lattice | tetragonal | tetragonal | tetragonal | tetragonal | tetragonal | tetragonal |
| space group | I4/mmm | I4/mmm | I4/mmm | I4/mmm | I4/mmm | I4/mmm |
| $a$ (Å) | 3.9175(3) | 3.9003(2) | 3.8957(2) | 3.8695(2) | 3.8570(3) | 3.844(2) |
| $c$ (Å) | 24.525(2) | 24.495(2) | 24.439(1) | 24.435(2) | 24.440(2) | 24.424(2) |
| $V$ (Å³) | 376.38(5) | 372.62(4) | 370.35(3) | 365.87(3) | 363.58(4) | 360.89(4) |
| $D$ (g cm⁻³) | 8.037 | 8.172 | 8.237 | 8.433 | 8.530 | 8.647 |
| Rp (%) | 7.837 | 7.058 | 5.390 | 7.534 | 8.004 | 7.666 |
| Rwp (%) | 7.632 | 5.957 | 4.746 | 5.911 | 6.518 | 6.331 |
| $\chi^2$ | 2.437 | 1.592 | 1.840 | 1.760 | 1.940 | 2.421 |

$R_p = \sum |y_{io} - y_{ic}| / \sum |y_{io}|$, $R_{wp} = \left[ \sum w_i (y_{io} - y_{ic})^2 / \sum w_i y_{io}^2 \right]^{1/2}$, $\chi^2 = \sum \left[ w_i (y_{io} - y_{ic})^2 / (N - P_1 - P_2) \right]$

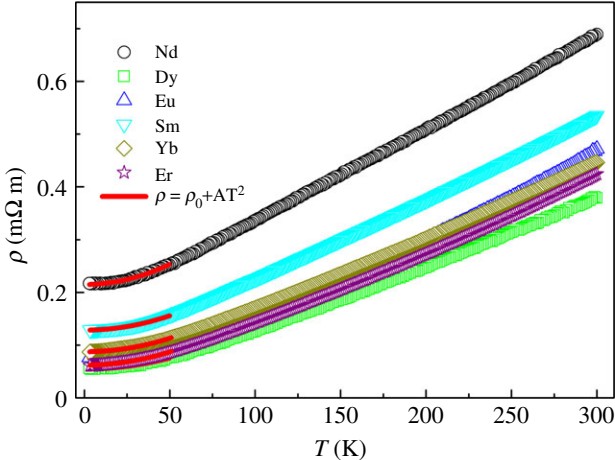

**Figure 3.** The temperature dependence of the resistivity of $Bi_2LnO_4Cu_2Se_2$ from 2 to 300 K. The red lines are the Fermi-liquid fitting of the resistivity at low temperature.

**Table 3.** Fractional atomic coordinates, bond length and bond angles determined by the refinement of X-ray diffraction data.

| formula | $Bi_2NdO_4Cu_2Se_2$ | $Bi_2SmO_4Cu_2Se_2$ | $Bi_2EuO_4Cu_2Se_2$ | $Bi_2DyO_4Cu_2Se_2$ | $Bi_2ErO_4Cu_2Se_2$ | $Bi_2YbO_4Cu_2Se_2$ |
|---|---|---|---|---|---|---|
| Z(Bi) | 0.8951(1) | 0.8960(1) | 0.8958(1) | 0.8973(2) | 0.8980(2) | 0.8987(1) |
| Z(Se) | 0.3097(3) | 0.3099(3) | 0.3111(2) | 0.3104(3) | 0.3100(3) | 0.3116(2) |
| Z(O) | 0.9427(9) | 0.9456(9) | 0.9442(6) | 0.947(1) | 0.948(1) | 0.9476(9) |
| $d_{(Bi-O)}$(Å) | 2.28(1) | 2.30(1) | 2.277(8) | 2.28(2) | 2.29(2) | 2.27(1) |
| $d_{(Bi-Se)}$(Å) | 3.472(4) | 3.472(4) | 3.445(3) | 3.462(5) | 3.474(5) | 3.453(4) |
| $d_{(Ln-O)}$(Å) | 2.41(1) | 2.36(1) | 2.377(9) | 2.30(2) | 2.30(2) | 2.30(1) |
| $d_{(Cu-Se)}$(Å) | 2.445(4) | 2.440(4) | 2.453(3) | 2.434(5) | 2.422(5) | 2.440(3) |
| $Angle_{(O-Bi-O)}$ | $118.5(9) \times 2$ | $116(1) \times 2$ | $117.5(7) \times 4$ | $116(1) \times 2$ | $115(2) \times 2$ | $116(1) \times 2$ |
| (degree) | $74.8(4) \times 4$ | $73.8(5) \times 4$ | $74.4(3) \times 4$ | $73.7(6) \times 4$ | $73.1(6) \times 4$ | $73.6(5) \times 4$ |
| $Angle_{(Se-Cu-Se)}$ | $106.5(3) \times 2$ | $106.1(3) \times 2$ | $105.1(2) \times 2$ | $105.3(3) \times 2$ | $105.5(3) \times 2$ | $104.0(2) \times 2$ |
| (degree) | $111.0 \times 4$ | $111.2(1) \times 4$ | $111.7(1) \times 4$ | $111.6(2) \times 4$ | $111.5(2) \times 4$ | $112.3(1) \times 4$ |

The magnetic behaviour of $Bi_2SmO_4Cu_2Se_2$ is not remarkable and the inverse susceptibility curve for $Bi_2SmO_4Cu_2Se_2$ is convex, indicating the itinerant electron susceptibility is non-negligible. We fitted the susceptibility curve in the whole measured temperature range using the modified Curie–Weiss law, $\chi = C/(T - \theta_P) + \chi_0$. We calculated the effective moment from the fitted parameters which are plotted in electronic supplementary material, figure S3. These values are all consistent with the expected values for the trivalent Ln free ions, which can be expressed as $\mu = g\sqrt{J(J+1)}$. All the Weiss constants are negative, suggesting antiferromagnetic exchange interactions, but further measurements are required for more detailed interpretation.

In order to verify the metallic ground state, we studied the density of states (DOS) and the band structure of $Bi_2NdO_4Cu_2Se_2$ calculated using DFT. The crystal structures of $Bi_2NdO_4Cu_2Se_2$ were optimized with respect to the lattice parameters and atomic positions. All the optimized lattice parameters and atomic coordinates are in good agreement with the experimental observation. The calculated DOS of $Bi_2NdO_4Cu_2Se_2$ is shown in figure 4a. There is finite DOS at the Fermi level, indicating the metallic ground state. The valence band maximum of $Bi_2NdO_4Cu_2Se_2$ mainly consists of Cu-3d and Se-4p electrons. The conduction band minimum is mainly built up of Bi-6p and O-2p electrons. Figure 4b shows the band structure of $Bi_2NdO_4Cu_2Se_2$. There are two bands crossing the $E_F$ and the $E_F$ locates at the valence band, indicating the metallic ground state and the hole-type carriers. There is a small gap between the valence band and conduction band at $\Gamma$ point. Other $Bi_2LnO_4Cu_2Se_2$ have similar band structures which are not shown here.

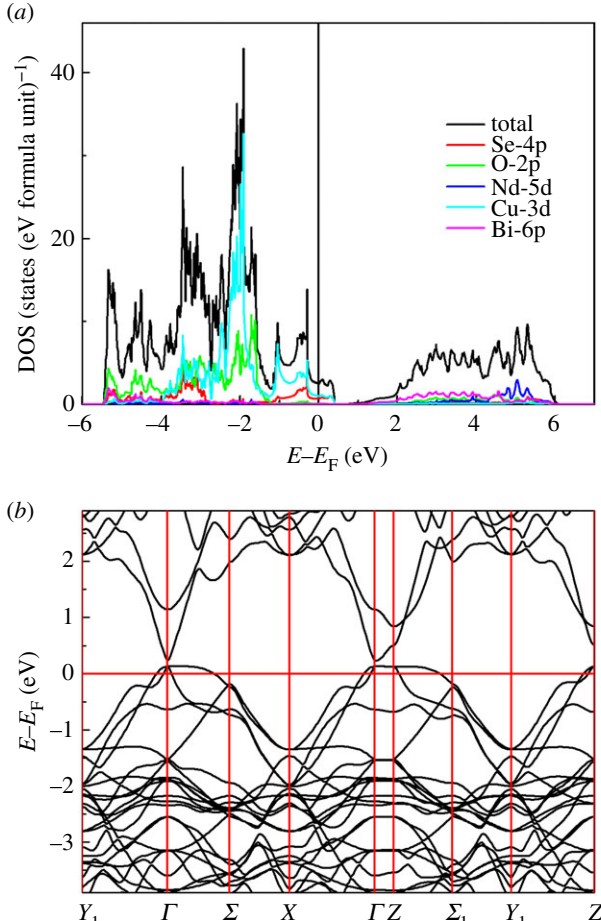

**Figure 4.** (*a*) The DOS and (*b*) the band structure of $Bi_2NdO_4Cu_2Se_2$.

## 4. Conclusion

In summary, we have synthesized a new series of layered oxyselenides $Bi_2LnO_4Cu_2Se_2$ through solid-state reaction. Their crystalline structures were studied through X-ray diffraction experiment. These compounds were crystalline in tetragonal structure with space group I4/mmm. All these compounds exhibit metallic transport property with hole-type carriers, which is verified by the DFT calculation. All the samples exhibit paramagnetic behaviour and no magnetic transition was found from 5 to 300 K. The new $Bi_2LnO_4$-type block in these compounds may give people some enlightenment in synthesizing new iron-based superconductors or other layered compounds.

Data accessibility. The datasets supporting this article have been uploaded as part of the electronic supplementary material.
Authors' contributions. S.T., C.G. and C.W. carried out the solid-state reaction experiment, participated in data analysis, participated in the design of the study and drafted the manuscript; Q.J., G.Y. and M.S. carried out the crystal structure and physical properties analyses and critically revised the manuscript; F.X. and R.C. collected field data and critically revised the manuscript; S.T. and Y.S. conceived of the study, designed the study, coordinated the study and helped draft the manuscript. All authors gave final approval for publication and agree to be held accountable for the work performed therein.
Competing interests. There are no conflicts to declare.
Funding. This work was supported by the National Nature Science Foundation of China under contract nos. 51802177 and 11804194, the Natural Science Foundation of Shandong Province under grant no. ZR2019MA020 and the Research Start-up Funds of Shandong University of Science and Technology (no. 415058).

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
