## [Reviewer comments · Royal Society Open Science]

Review History

RSOS-201078.R0 (Original submission)

Review form: Reviewer 1

Is the manuscript scientifically sound in its present form?

Yes

Are the interpretations and conclusions justified by the results?

Yes

Is the language acceptable?

Yes

Do you have any ethical concerns with this paper?

No

Have you any concerns about statistical analyses in this paper?

No

Recommendation?

Accept with minor revision (please list in comments)

Comments to the Author(s)

In this work, the authors have reported the preparations, structural, transport, and magnetic properties of layered oxyselenides $\text{Bi}_2\text{LnO}_4\text{Cu}_2\text{Se}_2$ ($\text{Ln} = \text{Nd, Sm, Eu, Dy, Er, Yb}$). The metallic ground states of those compounds are well explained by their DFT calculation. This work is complete and will be helpful for the experimental explorations of the new material systems with the novel quasi-2D layered structure. Therefore, I recommend it to be published, after some minor comments below have been considered:

1. The authors have analyzed the structural properties of the $\text{Bi}_2\text{LnO}_4\text{Cu}_2\text{Se}_2$ for different Ln. However, the influence of the different Ln atoms on the electric transport properties is not clear. I suggest the authors add some discussion addressing this point.
2. Some equations, such as that below Table 2, are not displayed correctly. Please fix them.
3. It is difficult to distinguish the symbols in Fig. 3.

Review form: Reviewer 2

Is the manuscript scientifically sound in its present form?

No

Are the interpretations and conclusions justified by the results?

No

Is the language acceptable?

No

Do you have any ethical concerns with this paper?

No

Have you any concerns about statistical analyses in this paper?

No

Recommendation?

Reject

Comments to the Author(s)

This paper describes several lanthanide bismuth copper oxide selenides. In principle the work is satisfactory, although it does not offer much insight beyond that of the Evans paper from several years ago.

The samples and the structural analysis seem fine.

The conductivity data and the band structure calculations show the expected metallic behaviour because of holes in the valence band.

The magnetic measurements could go in the supporting information. These are standard data for the lanthanides. The Sm compound has been wrongly interpreted - the curvature is because there are two J states involved. The data could likely be fitted by the Hamaker equation (although the value in doing this is limited).

The comment on the Cu-Se bond lengths is confusing and the more I read it the less it makes sense. In these compounds the states at the top of the valence band are oxidised leading to the metallic properties. These states are a mixture of $\text{Cu}3d$ and $\text{Se}4p$ states of very similar energy.

They are also antibonding, so depletion of them leads to shortening of the Cu-Se lengths compared with LaOCuSe. Chemically it's probably best to think of Cu⁺ and selenide (Se²⁻) both being partially oxidised.

The description of the structure on p2 is a bit longwinded. It's probably best to describe the oxide part as a "fluorite-type slab"

There is rather a lot on iron-based superconductors and superconductivity in the introduction and elsewhere. I think just a comment that many of the compounds with CuSe layers have analogues with FeAs layers, and there could be iron arsenides with this structure would be sufficient.

Overall I think this needs to be a resubmission as there are some errors of interpretation.

Decision letter (RSOS-201078.R0)

Dear Dr Tan:

Title: Synthesize, structure and physical properties of the new layered oxyselenides Bi₂LnO₄Cu₂Se₂ (Ln = rare earth)
Manuscript ID: RSOS-201078

The editor assigned to your manuscript has now received comments from reviewers. We would like you to revise your paper in accordance with the referee and Subject Editor suggestions which can be found below (not including confidential reports to the Editor). Please note this decision does not guarantee eventual acceptance.

Please submit your revised paper before 27-Aug-2020. Please note that the revision deadline will expire at 00.00am on this date. If we do not hear from you within this time then it will be assumed that the paper has been withdrawn. In exceptional circumstances, extensions may be possible if agreed with the Editorial Office in advance. We do not allow multiple rounds of revision so we urge you to make every effort to fully address all of the comments at this stage. If deemed necessary by the Editors, your manuscript will be sent back to one or more of the original reviewers for assessment. If the original reviewers are not available we may invite new reviewers.

On behalf of the Subject Editor Professor Anthony Stace and the Associate Editor Professor Tobias Hertel.

RSC Associate Editor:
Comments to the Author:
(There are no comments.)

RSC Subject Editor:
Comments to the Author:
(There are no comments.)

Reviewers' Comments to Author:
Reviewer: 1

Comments to the Author(s)

In this work, the authors have reported the preparations, structural, transport, and magnetic properties of layered oxyselenides $\text{Bi}_2\text{LnO}_4\text{Cu}_2\text{Se}_2$ (Ln = Nd, Sm, Eu, Dy, Er, Yb). The metallic ground states of those compounds are well explained by their DFT calculation. This work is complete and will be helpful for the experimental explorations of the new material systems with the novel quasi-2D layered structure. Therefore, I recommend it to be published, after some minor comments below have been considered:

1. The authors have analyzed the structural properties of the $\text{Bi}_2\text{LnO}_4\text{Cu}_2\text{Se}_2$ for different Ln. However, the influence of the different Ln atoms on the electric transport properties is not clear. I suggest the authors add some discussion addressing this point.
2. Some equations, such as that below Table2, are not displayed correctly. Please fix them.
3. It is difficult to distinguish the symbols in Fig. 3.

Reviewer: 2

Comments to the Author(s)

This paper describes several lanthanide bismuth copper oxide selenides. In principle the work is satisfactory, although it does not offer much insight beyond that of the Evans paper from several years ago.

The samples and the structural analysis seem fine.

The conductivity data and the band structure calculations show the expected metallic behaviour because of holes in the valence band.

The magnetic measurements could go in the supporting information. These are standard data for the lanthanides. The Sm compound has been wrongly interpreted - the curvature is because there are two J states involved. The data could likely be fitted by the Hamaker equation (although the value in doing this is limited).

The comment on the Cu-Se bond lengths is confusing and the more I read it the less it makes sense. In these compounds the states at the top of the valence band are oxidised leading to the metallic properties. These states are a mixture of Cu^{3d} and Se 4p states of very similar energy. They are also antibonding, so depletion of them leads to shortening of the Cu-Se lengths compared with LaOCuSe. Chemically it's probably best to think of Cu⁺ and selenide (Se²⁻) both being partially oxidised.

The description of the structure on p2 is a bit longwinded. It's probably best to describe the oxide part as a "fluorite-type slab"

There is rather a lot on iron-based superconductors and superconductivity in the introduction and elsewhere. I think just a comment that many of the compounds with CuSe layers have analogues with FeAs layers, and there could be iron arsenides with this structure would be sufficient.

Overall I think this needs to be a resubmission as there are some errors of interpretation.

Author's Response to Decision Letter for (RSOS-201078.R0)

See Appendix A.

Decision letter (RSOS-201078.R1)

Dear Dr Tan:

Title: Synthesize, structure and physical properties of the new layered oxyselenides

Bi₂LnO₄Cu₂Se₂ (Ln = rare earth)

Manuscript ID: RSOS-201078.R1

It is a pleasure to accept your manuscript in its current form for publication in Royal Society Open Science. The chemistry content of Royal Society Open Science is published in collaboration with the Royal Society of Chemistry.

On behalf of the Subject Editor Professor Anthony Stace and the Associate Editor Professor Tobias Hertel.

RSC Associate Editor
Comments to the Author:
(There are no comments.)

Reviewer(s)' Comments to Author:

Appendix A

Dear Editor,

Thanks for your email concerning our manuscript entitled “**Synthesize, structure and physical properties of the new layered oxyselenides $\text{Bi}_2\text{LnO}_4\text{Cu}_2\text{Se}_2$ ($\text{Ln} = \text{rare earth}$)**” (RSOS-201078) together with the comments of referees. We thank the referees for their comments and suggestions. All the comments are responded carefully, as shown following:

Reviewers' Comments to Author:

Reviewer: 1

Comments to the Author(s)

In this work, the authors have reported the preparations, structural, transport, and magnetic properties of layered oxyselenides $\text{Bi}_2\text{LnO}_4\text{Cu}_2\text{Se}_2$ ($\text{Ln} = \text{Nd, Sm, Eu, Dy, Er, Yb}$). The metallic ground states of those compounds are well explained by their DFT calculation. This work is complete and will be helpful for the experimental explorations of the new material systems with the novel quasi-2D layered structure. Therefore, I recommend it to be published, after some minor comments below have been considered:

1. The authors have analyzed the structural properties of the $\text{Bi}_2\text{LnO}_4\text{Cu}_2\text{Se}_2$ for different Ln. However, the influence of the different Ln atoms on the electric transport properties is not clear. I suggest the authors add some discussion addressing this point.

Response: Thanks for the comment. As the DOS at the Fermi level mainly originates from Cu 3d and Se 4p orbitals. The Ln atoms contributes little to the DOS and so to the transport properties. In fact, the influence of Ln atoms to the transport properties is mainly embodied in the influence of the structure of the Cu-Se layer. As the Ln atom changed from Nd to Yb, the size of which is decreasing, the angle of Se-Cu-Se and the distance of Cu-Cu tend to decrease. As a result, the electrons transferred easily in Cu-Se layers. This can be demonstrated by the decreasing resistivity and smaller temperature coefficient of resistivity as Ln changing from Nd to Yb. We have added the related explanations in the manuscript.

2. Some equations, such as that below Table2, are not displayed correctly. Please fix them.

Response: Thanks for the comment. We have corrected it.

3. It is difficult to distinguish the symbols in Fig. 3.

Response: Thanks for the comment. We have optimized the figure.

Reviewer: 2

Comments to the Author(s)

This paper describes several lanthanide bismuth copper oxide selenides. In principle the work is satisfactory, although it does not offer much insight beyond that of the Evans paper from several years ago.

The samples and the structural analysis seem fine.

The conductivity data and the band structure calculations show the expected metallic behaviour because of holes in the valence band.

The magnetic measurements could go in the supporting information. These are standard data for the lanthanides. The Sm compound has been wrongly interpreted - the curvature is because there are two J states involved. The data could likely be fitted by the Hamaker equation (although the value in doing this is limited).

Response: Thanks for the comment. We have moved the magnetic measurements (Figure 4) into the supporting information (Fig. S3 in ESI). According to the comment, which recommended us to fit the magnetic data of Sm compound through the Hamaker equation, we tried to investigate the Hamaker equation. However, we found the Hamaker theory is related to the van der Waals attraction between spherical particles. We do not understand the relationship between them and how to fit by the equation. We are grateful if the referee could provide us some references.

The comment on the Cu-Se bond lengths is confusing and the more I read it the less it makes sense. In these compounds the states at the top of the valence band are oxidised leading to the metallic properties. These states are a mixture of Cu 3d and Se 4p states of very similar energy. They are also antibonding, so depletion of them leads to shortening of the Cu-Se lengths compared with LaOCuSe. Chemically it's probably best to think of Cu⁺ and selenide (Se²⁻) both being partially oxidised.

Response: Thanks for the comment. We agree. We just want to prove the different state of CuSe through comparing the bond length of Cu-Se between Bi₂LnO₄Cu₂Se₂ and La(Bi)OCuSe. In La(Bi)OCuSe, the valence state of Cu is +1 (It may be not suitable to discuss the valence state of Cu because they are antibonding.) While in Bi₂LnO₄Cu₂Se₂, the valence state of Cu is about +1.5 (or a mixture of Cu⁺/Cu²⁺). The difference can be reflected by the bond length of Cu-Se. In other words,

as you stated, the CuSe is being partially oxidized in $\text{Bi}_2\text{LnO}_4\text{Cu}_2\text{Se}_2$ compared with LaOCuSe or BiOCuSe.

The description of the structure on p2 is a bit longwinded. It's probably best to describe the oxide part as a "fluorite-type slab"

Response: Thanks for the comment. We agree. We have corrected it in the manuscript.

There is rather a lot on iron-based superconductors and superconductivity in the introduction and elsewhere. I think just a comment that many of the compounds with CuSe layers have analogues with FeAs layers, and there could be iron arsenides with this structure would be sufficient.

Response: Thanks for the comment. We agree. We have deleted the related comment about iron-based superconductors and added the recommended comment in our manuscript.

Finally, we would like to thank referees for improving our manuscript with valuable advices. We do hope that the revised manuscript can satisfy referees' requirements and be considered for publication in *Royal Society Open Science*.

Sincerely on behalf of the authors,

Shugang Tan